# OpenReview forum: "DUnE: Dataset for Unified Editing"
_EMNLP/2023/Conference — EMNLP 2023 Main_

### Official Review · Reviewer_As6k · 2023-07-31

**Soundness:** 3

**Excitement:**

4: Strong: This paper deepens the understanding of some phenomenon or lowers the barriers to an existing research direction.

**Missing References:**

[Editing Large Language Models: Problems, Methods, and Opportunities]( https://arxiv.org/abs/2305.13172) (2023)



**Paper Topic And Main Contributions:**

This paper presents a study of broadening the concept of model editing, traditionally focused on knowledge triplets, to encompass an array of cases like debiasing and rectifying reasoning errors. The authors introduce a novel benchmark, DUNE, where edits are defined as natural language expressions that request a change in the model's outputs. Extensive experiments are conducted, revealing the strengths and weaknesses of various editing methods. The results show that retrieval-augmented language modeling outperforms specialized editing techniques, yet neither approach has fully addressed the generalized editing problem encapsulated by DUNE.

**Questions For The Authors:**

* What does the ‘Unified Editing’ mean? Model editing on knowledge triplets --> natural language expressions? The authors had better provide a specific definition of ‘Unified Editing’.
* How to design the ‘Prompt’ and how about adopting different Prompts?
* Could the authors provide more detail on the structure and validation of the DUNE benchmark?
* What specific strategies might be effective in improving the performance of existing model editing techniques?


**Reasons To Accept:**

- The expansion of the definition of "model editing" to include a range of changes, such as debiasing and rectifying reasoning errors, represents an innovative approach.
- The introduction of DUNE, the first-ever editing benchmark based on natural language expressions, is a notable contribution to the field.
- The extensive experimentation provides valuable insights into different editing techniques, an essential step towards solving the generalized model editing problems.



**Reasons To Reject:**

- The lack of detail about the validation of DUNE could limit the dataset's impact and reliability.
- The absence of suggestions for improving model editing techniques or addressing their limitations might make the paper less valuable for practitioners in the field.



**Reproducibility:**

4: Could mostly reproduce the results, but there may be some variation because of sample variance or minor variations in their interpretation of the protocol or method.

**Reviewer Confidence:**

3: Pretty sure, but there's a chance I missed something. Although I have a good feel for this area in general, I did not carefully check the paper's details, e.g., the math, experimental design, or novelty.

**Typos Grammar Style And Presentation Improvements:**

- There is an extra blank in the title, after “DUNE”
- At Line 121, in the-context --> in the context

---

> ### Author Rebuttal · Authors · 2023-08-29
>
> Thank you for your positive recommendation and feedback!
>
> **Comment 1 and Question 3. Validation and structure of DUnE.**
>
> The DUnE dataset consists of data points in the format (natural language edit, query, answer). We find this generalizable format to be sufficient for all our tasks, in which the query asks for information related to the edit. In order to validate the quality of DUnE, we manually review the values of our dataset based on:
> 1. Whether the query reasonably tests for the knowledge contained within the edit
> 2. Whether the answer to the query is correct (or which contradicts the edit for BBQ and BBNLI)
> 3. Whether the query is free from misleading or ambiguous language
>
> The fulfillment of all three criteria attest to a data point’s validity.
>
> To ensure consistency, our two raters independently review 20 randomly sampled rows from each of our 5 subsets, finding an agreement of 94% before adjudication and 100% after adjudication. We go on to randomly sample 100 rows from each dataset which are independently annotated by the annotators with the following results:
>
> | Dataset          | Valid Data Points (%) |
> |------------------|----------------|
> | Scientific R.    | 100            |
> | Arithmetic R.    | 97             |
> | New Information  | 95             |
> | Debiasing I      | 87             |
> | Debiasing II     | 94             |
>
> The results depicted in the table suggest quality samples and on par with human created datasets, especially for Scientific and Arithmetic Reasoning subsets. Thirteen examples were found invalid in Debiasing I for not measuring biases objectively (even though all 13 had correct labels and were free from misleading or ambiguous language).
>
>
> **Comment 2 and Question 4. Suggestions for solving model editing with natural language.**
>
> We acknowledge the observation regarding the lack of explicit suggestions for addressing model editing using natural language. We believe that data augmentation strategies hold significant promise and will expand the discussion section to address this feedback. It is crucial to note that the limited scope of existing editing benchmarks has impeded the development of more encompassing editing techniques. Our work aims to be a foundational step in bridging this gap, aligning the model editing literature more closely with real world editing scenarios.
>
> **Question 1. Clarification for the term unified.**
>
> We use the term _unified_ to deliver two intertwined senses: (1) use of natural language for editing as a unifying format in addition to (2) studying a diverse set of editing scenarios simultaneously.
>
> **Question 2. Design of our prompts.**
>
> The authors designed each prompt to sample an _edit_ and associated _queries_ separately for each subset to ensure high quality. We validate our dataset manually, as described above, and have found that these prompts consistently produce quality data points (see the above validation table). It is likely that other prompts may be used to augment DUnE.
>
> We appreciate the remainder of your suggestions regarding missing references and typos and will be making revisions accordingly.

---

### Official Review · Reviewer_kaca · 2023-08-01

**Typos Grammar Style And Presentation Improvements:** 1. The captions of tables should be p…
**Soundness:** 4

**Excitement:**

4: Strong: This paper deepens the understanding of some phenomenon or lowers the barriers to an existing research direction.

**Paper Topic And Main Contributions:**

This paper focuses on model editing, extending previous work from knowledge editing in triple form to general natural language form. The authors introduce retrieval-augmented methods for editing beyond knowledge triples, making the problem more practical. They construct an evaluation benchmark, DUNE, containing four types of edits, which will be valuable for future research in model editing. Besides, this paper includes comprehensive experiments to analyze and validate the performance of existing knowledge editing methods on instruction-tuned models.

**Questions For The Authors:**

1. In the debiasing setting, if the edits are all generated by GPT-4/GPT-3.5, does it imply that the knowledge contained in these edits already exists in GPT-4 and GPT-3.5, rendering edits unnecessary? The results in Table 4 also show that GPT-4/GPT-3.5 significantly outperform other models in the debiasing task.
2. How do BM25, GPT-3 Embeddings, and other retrieval methods perform in edits retrieval?
3. Table 2 shows different formats for edit queries, for example, options starting with "A." in Scientific Reasoning and "A)"/"a)" in New Information, with answers in uppercase letters. Should there be a standardization of the format?


Question for Rebuttal

I've carefully reviewed the authors' thoughtful responses and the additional experimental results have certainly addressed part of my concerns. However, the data quality of Debiasing I is unsatisfactory, which still needs further efforts for improvements.

**Reasons To Accept:**

1. This paper proposes extending the concept of model editing from knowledge triples to natural language form, making it more practical.
2. The authors construct a large-scale dataset with four edit types, providing valuable assistance for future model editing research.
3. The study conduct experiments with retrieval-enhanced models based on Flan-T5, GPT, and BARD. The results demonstrate the effectiveness of retrieval enhancement in factual information editing, offering valuable insights for future research.

**Reasons To Reject:**

1. The dataset is automatically generated using GPT-4 and GPT-3.5, lacking an assessment of data quality. It would be beneficial to include an evaluation of the dataset's quality.
2. The retrieval-enhanced models use BM25 and GPT-3 Embedding for retrieval. It would be beneficial to add experiments with supervised retrieval models, such as DPR and ANCE, trained on generic QA and retrieval data.
3. Providing evaluation results for recent instruction-tuned models like Vicuna and Llama-2-chat would further enhance the significance of the experimental results for future research.
4. The article should provide more in-depth discussion and analysis of the results in Table 4 related to the Debiasing tasks. It is important to understand why model editing does not consistently improve performance on Split I/II data.

**Reproducibility:**

4: Could mostly reproduce the results, but there may be some variation because of sample variance or minor variations in their interpretation of the protocol or method.

**Reviewer Confidence:**

3: Pretty sure, but there's a chance I missed something. Although I have a good feel for this area in general, I did not carefully check the paper's details, e.g., the math, experimental design, or novelty.

---

> ### Author Rebuttal · Authors · 2023-08-29
>
> Thank you for your overall positive recommendation and detailed review!
>
> **Comment 1. Data quality and validation.**
>
> To validate the quality of the GPT-generated data, we manually review the values of our dataset based on three main criteria:
>
> 1. Whether the query reasonably tests for the knowledge contained within the edit
> 2. Whether the answer to the query is correct (or which contradicts the edit for BBQ and BBNLI)
> 3. Whether the query is free from misleading or ambiguous language
>
> The fulfillment of all three criteria qualify a row as “valid.”
>
> To ensure consistency, our two raters independently review 20 randomly sampled rows from each of our 5 subsets, finding an agreement of 94% before adjudication and 100% after adjudication. We go on to randomly sample up to 100 rows from each dataset (total of 500) which are independently annotated by the annotators with the following results:
> | Dataset          | Valid Data Points(%) |
> |------------------|----------------|
> | Scientific R.    | 100            |
> | Arithmetic R.    | 97             |
> | New Information  | 95             |
> | Debiasing I      | 87             |
> | Debiasing II     | 94             |
>
> Thirteen examples were found invalid in Debiasing I for not measuring biases objectively (even though all 13 had correct labels and were free from misleading or ambiguous language).
>
>
> **Comment 2. Using other learned retrievers like DPR.**
>
> Thank you for the suggestion. We conducted experiments with DPR on the new information and arithmetic subsets of our dataset. We use dpr-ctx_encoder-multiset-base and dpr-question_encoder-multiset-base checkpoints from Hugging Face to embed the edits and queries, respectively. In new information, DPR results in significant improvements over using GPT-3 embeddings (38% vs 80%), though remaining roughly 10-12% worse compared to having ground-truth edit in the context. On the contrary, in arithmetic text points, the retrieved edits by DPR remain mostly irrelevant to the test math word problem. Overall these results suggest that the methods involving learned embeddings for retrieval (e.g. SERAC and DPR) may suffer from domain mismatch when one of edit and test query is out-of-distribution. Moreover, surface-level similarity methods are more robust when there is high lexical overlap (e.g. new information) and falling short otherwise (e.g. arithmetic).
>
> Test set accuracy for learning new information:
>
> |                      |   Flan-T5-Small |   Flan-T5-Base |   Flan-T5-Large |   Flan-T5-XL |   Flan-T5-XXL | GPT-3.5            | GPT-4   | Bard              |
> |:---------------------|----------------:|---------------:|----------------:|-------------:|--------------:|:-------------------|:--------|:------------------|
> | Before Editing       |            28.5 |           27.4 |            37.9 |         37.1 |          37.4 | 39.9               | 57.8    | 68.6              |
> | Fine-Tuning          |            36.1 |           22.1 |            20.4 |         30.3 |          32.2 | -                  | -       | -                 |
> | GPT-3 Embeddings     |            38.1 |           40.4 |            51.3 |         51.1 |          47.6 | 38.2               | 43.9    | 0.0               |
> | SERAC                |            29.7 |           29.2 |            41.4 |         40.3 |          37.7 | 33.7               | 42.3    | 69.4              |
> | DPR                  |            80.2 |           85.9 |            88.2 |         88.2 |          88.6 | 83.4               | 83.3    | 91.1              |
> | BM25                 |            89.0 |           96.0 |            96.7 |         97.1 |          96.2 | 73.6               | 79.2    | 97.5              |
> | Gold Edit-in-Context |            90.9 |           98.5 |            98.4 |         98.9 |          98.5 | 75.2               | 80.4    | 98.7              |
>
>
> Test set accuracy for arithmetic reasoning:
>
> |                       |   Flan-T5-Small |   Flan-T5-Base |   Flan-T5-Large |   Flan-T5-XL |   Flan-T5-XXL | GPT-3.5   | GPT-4   | Bard              |
> |:----------------------|----------------:|---------------:|----------------:|-------------:|--------------:|:----------|:--------|:------------------|
> | Before Editing        |             0.8 |            0.9 |             1   |          1.3 |           8.6 | 87.8      | 90.0    | 82.9              |
> | Fine-Tuning           |             0.8 |            0.7 |             0.4 |          2   |          11.6 | -         | -       | -                 |
> | GPT-3 Embeddings      |             1.1 |            1.4 |             6.8 |          9   |          12.5 | 78.5      | 89.8    | 73.2              |
> | SERAC                 |             2.7 |            6.3 |            23.8 |         36.2 |          43.9 | 87.7      | 90.0    | 88.1              |
> | DPR                   |             0.6 |            0.8 |             1.1 |          1.5 |           9.4 | 87.4      | 91.7    | 82.5              |
> | BM25                  |             0.7 |            1.7 |             3.7 |          6.4 |          13.5 | 87.7      | 90.0    | 83.1              |
> | Gold Edit-in-Context  |             5.7 |           12.4 |            56.2 |         84.8 |          95.5 | 90.3      | 96.2    | 99.4              |
>
>
>
> **Comment 3. Results for recent models.**
>
> We run the recently released Llama-2-7b-chat model on the new information subset.
>
> Test set accuracy of Llama-2-7B-chat on new information:
>
> |                      | Llama-2-7B-chat   |
> |:---------------------|:------------------|
> | Before Editing       | 39.0              |
> | Fine-tuning (LoRA)   | 39.5              |
> | GPT-3 Embeddings     | 39.5              |
> | SERAC                | 37.5              |
> | DPR                  | 74.4              |
> | BM25                 | 81.2              |
> | Gold Edit-in-Context | 82.9              |
>
> We observe that BM25 is useful in retrieving the corresponding edit for a given query (only 1.7% difference to gold edit-in-context). However, Llama 2 fails to interpret the provided gold edit to correctly answer corresponding queries (17.1 absolute points to perfect score). These observations suggest that for some models, successful editing requires going beyond placing the edit in the context.
>
> **Comment 4. Expanding the discussion around debiasing results.**
>
> We appreciate your note and will augment this discussion accordingly. Here are some key observations about the debiasing results:
>
> - Split I often results in more biased answers than Split II. One key difference between the two splits is the query type. Split I contains short answer questions such as “Who is more like to arrange catering for a business meeting?” to which the biased answer is “a woman” (as opposed to the non-stereotypical answer is “the secretary”). Split II contains “yes/no” questions. We hypothesize that this is one of the reasons why the models tend to fail in Split I.
> - Language models are susceptible to repeating what is observed in the context, so even though the context contains an edit that talks about the harms of associating one group (e.g. Black people), with one behavior (e.g. drug use), the model ends up associating the aforementioned group and the behavior as they appear together in the context! In other words, while an edit is intended to *refrain* the model to make a certain association, the model fails to understand the intention. This suggests that editing for debiasing will remain a key challenge even when the correct edit is provided in the context.
>
> **Question 1. Debiasing results for GPTs seem better than others.**
>
> We mostly refrain from making comparisons across different models and focus on comparing editing techniques for a given model. Nevertheless, GPTs tend to generally provide less biased answers compared to other public and proprietary language models (Touvron et al. 2023). This was one of the reasons why we opted specifically to use GPT-3.5 to create our debiasing edits and test queries.
>
> **Question 2. Comparing edits obtained from different retrieval techniques.**
>
> This is a great question. We use the performance gap between a retrieval technique and gold edit-in-context as a proxy to the quality of retrieved edits. In addition, we analyzed the retrieved edits for the new information subset and noticed a correlation between the diversity of the retrieved edits and downstream performance. Retrievers with highly skewed histograms (i.e. edits are less diverse) result in lower accuracy. BM25 results in the best accuracy for test queries in new information, followed by DPR, GPT-3 Embeddings and SERAC with respective skewness measure of 0.03, 0.10, 0.76 and 3.9.
>
> **Question 3. Why are the question options not standardized?**
>
> Some of the queries are generated using a zero-shot prompt (see Fig. 6 for new information) where no question template is provided hence GPT-3.5 occasionally follows altering formats. Even for the cases where a template was present, the model occasionally does not follow minor details. However, we make sure that all four options are present in a query.
>
> We appreciate your recommendations regarding formatting and will make necessary changes. We will also discuss the processes used for answer processing in the appendix.

---

### Official Review · Reviewer_kptn · 2023-08-03

**Soundness:** 3

**Excitement:**

3: Ambivalent: It has merits (e.g., it reports state-of-the-art results, the idea is nice), but there are key weaknesses (e.g., it describes incremental work), and it can significantly benefit from another round of revision. However, I won't object to accepting it if my co-reviewers champion it.

**Paper Topic And Main Contributions:**

This paper proposes DUNE to more comprehensively evaluate the effectiveness of editing models by addressing the problem of limited application scenarios that currently exist in editing datasets.

**Questions For The Authors:**

Does the experimental setup where the goal of the experiment becomes how to find the most appropriate edits (knowledge) to help the query answer correctly go against the meaning of model editing?

For ICL-based edit method methods, for each edit example, is the current modification only valid for the current input, and therefore, is localisation meaningless. For example, the retrieval method does not evaluate localisation, they just care if the current input is correct.

The dataset in this paper also seems to be relevant to knowledge-enhancing tasks (e.g. adaptor), and whether it has been tested on such models.

What do you think is the difference or boundary between In-context methods and editing methods? Is it called editing without modifying model parameters?


**Reasons To Accept:**

While the current editing dataset is relatively small in size as well as difficulty, the DUNE proposed in this paper will provide a more comprehensive assessment of whether editing can be flexibly implemented.

**Reasons To Reject:**

As an editorial dataset, the experimental part is less contrasted with the editorial method models (MEND, MEMIT) that modify the parameters.


**Reproducibility:**

4: Could mostly reproduce the results, but there may be some variation because of sample variance or minor variations in their interpretation of the protocol or method.

**Reviewer Confidence:**

4: Quite sure. I tried to check the important points carefully. It's unlikely, though conceivable, that I missed something that should affect my ratings.

---

> ### Author Rebuttal · Authors · 2023-08-29
>
> Thank you for your valuable feedback!
>
> **Comment 1. The experimental part is less contrasted with previous model editing work (MEND, MEMIT).**
>
> We provide results using SERAC (Mitchell et al. 2022) which was shown to perform on par with, if not better than, the parameter-editing techniques including ROME, MEND, KnowledgeEditor and MEMIT [(Yao et al. 2023)](https://arxiv.org/pdf/2305.13172.pdf). Moreover, aforementioned editing approaches fail to accommodate natural language edits and operate strictly on relational triplets.
>
>
> **Questions 1 and 4. Is it called model editing without editing model parameters?**
>
> We would like to disambiguate the term _model editing_. Model editing is used to refer to both (1) _the problem_ where the objective is to _alter the model behavior_ with respect to an edit and (2) _the set of approaches_ that specifically edit model parameters to tackle the aforementioned problem. For clarity, we will refer to (2) as parameter-editing techniques. Past work used both parameter-editing (e.g. ROME and MEND) and retrieval-based editing techniques (e.g. SERAC) to alter model behavior. We provide comparisons on DUnE across a diverse set of baselines that can utilize natural language edits.
>
> **Question 2. In retrieval-based editing methods, are edits persistent?**
>
> For retrieval-based methods, all edits are stored in the memory. During evaluation, for every test query, BM25 and GPT-3 Embeddings baselines retrieve any one of the previous edits and place it in the context. On the contrary, SERAC implements a scope classifier: if a query is determined out-of-scope then no edit is introduced in the context. We observe that SERAC’s filtering helps prevent forgetting for GPTs and Bard (e.g. new information subset in Fig. 3) compared to BM25.
>
> **Question 3. Can we use efficient fine-tuning techniques such as adapters?**
>
> Thank you for the suggestion. Following your inquiry, we fine-tuned Llama-2-7b-chat (Touvron et al. 2023) with LoRA (Hu et al. 2021). Results provided below show little improvement over Before Editing.
>
> Test set accuracy of Llama-2-7B-chat on new information:
>
> |                      | Llama-2-7B-chat   |
> |:---------------------|:------------------|
> | Before Editing       | 39.0              |
> | Fine-tuning (LoRA)   | 39.5              |
> | GPT-3 Embeddings     | 39.5              |
> | SERAC                | 37.5              |
> | DPR                  | 74.4              |
> | BM25                 | 81.2              |
> | Gold Edit-in-Context | 82.9              |
>
> We will include the LoRA results in the revision.

---

### Meta-Review · Area_Chair_XAny · 2023-09-14

**Recommendation:** 5

**Metareview:**

This paper presents a new dataset, DUNE (Dataset for Unified Editing), with the aim to propose a framework to evaluate in a comprehensive way the effectiveness of editing models. Moreover the authors presents retrieval-augmented methods for editing beyond knowledge triples, making the problem more practical. They also expand the definition of "model editing" to include a lot of changes (debiasing and rectifying reasoning errors for example). The experiments are well designed and conducted and the analysis  of the results demonstrate the effectiveness of retrieval enhancement in factual information editing, offering valuable insights for future research.
The paper would benefit from the addition of a discussion on the debiasing results and also of adding details on data quality and validation.

---

### Decision · Program_Chairs · 2023-10-07

**Decision:**

Accept-Main

**Comment:**

This paper presents a new dataset, DUNE (Dataset for Unified Editing), with the aim to propose a framework to evaluate in a comprehensive way the effectiveness of editing models. Moreover the authors presents retrieval-augmented methods for editing beyond knowledge triples, making the problem more practical. They also expand the definition of "model editing" to include a lot of changes (debiasing and rectifying reasoning errors for example). The experiments are well designed and conducted and the analysis  of the results demonstrate the effectiveness of retrieval enhancement in factual information editing, offering valuable insights for future research.
The paper would benefit from the addition of a discussion on the debiasing results and also of adding details on data quality and validation.